# How Is Privacy Behavior Formulated? A Review of Current Research and Synthesis of Information Privacy Behavioral Factors

**Ioannis Paspatis [1,\*], Aggeliki Tsohou [1] and Spyros Kokolakis [2]**

[1] Department of Informatics, Ionian University, 49100 Corfu, Greece; atsohou@ionio.gr
[2] Department of Information and Communication Systems Engineering, University of Aegean, 83200 Samos, Greece; sak@aegean.gr
\* Correspondence: ipaspatis@ionio.gr; Tel.: +30-26610-87760

**Abstract:** What influences Information Communications and Technology (ICT) users' privacy behavior? Several studies have shown that users state to care about their personal data. Contrary to that though, they perform unsafe privacy actions, such as ignoring to configure privacy settings. In this research, we present the results of an in-depth literature review on the factors affecting privacy behavior. We seek to investigate the underlying factors that influence individuals' privacy-conscious behavior in the digital domain, as well as effective interventions to promote such behavior. Privacy decisions regarding the disclosure of personal information may have negative consequences on individuals' lives, such as becoming a victim of identity theft, impersonation, etc. Moreover, third parties may exploit this information for their own benefit, such as targeted advertising practices. By identifying the factors that may affect SNS users' privacy awareness, we can assist in creating methods for effective privacy protection and/or user-centered design. Examining the results of several research studies, we found evidence that privacy behavior is affected by a variety of factors, including individual ones (e.g., demographics) and contextual ones (e.g., financial exchanges). We synthesize a framework that aggregates the scattered factors that have been found in the literature to affect privacy behavior. Our framework can be beneficial to academics and practitioners in the private and public sectors. For example, academics can utilize our findings to create specialized information privacy courses and theoretical or laboratory modules.

**Keywords:** privacy behavior; determinant factors

## 1. Introduction

The academic community has demonstrated through various studies that internet users state to care about their personal data [1–4]. However, if this is true, it raises the question of why do they behave as if they are not interested in their protection? For example, they disclose vast amounts of personal information when using their preferred social network [5,6]). Some researchers have attempted to explain this contradiction using the term "the privacy paradox", pointing out the difference between privacy attitude and actual privacy behavior. Other researchers aired behavioral and psychological factors behind users' privacy behavior [1–4]. The term information privacy behavior refers to users' actions with regard to the protection of one's own privacy when using Information Communications and Technologies (ICTs), for example, as personal information disclosure, the application of controls to protect own personal data, or the configuration of privacy settings. Many researchers highlight that research in the domain of privacy behavior is challenging, mostly because it mainly relies on self-reported data from the participants in empirical investigations.

The "intrusion" of ICT in our lives and the growth of technology has put our life on the front scene of a global theater. We expose ourselves everywhere and all the time. We

are continuously connected to the internet through smartphones. Our internet-connected wearables are constantly measuring our daily activities and health condition. We post our moments and thoughts on Online Social Networks or Social Network Sites (OSNs or SNS) while the Internet of Things (IoT) play a "smarter" role in our lives day by day [7]. Privacy awareness among SNS users is a fundamental aspect of privacy behavior, as it influences their decisions regarding the disclosure of personal information on social networks. Self-disclosure of information may have negative consequences on individuals' lives, such as becoming a victim of identity theft, impersonation, etc. Moreover, third parties may exploit this information for their own benefit, such as targeted advertising practices [8,9] or affecting people's voting opinion, as Cambridge Analytica's scandal showed us [10]. By recognizing the reasons that can influence the privacy awareness of SNS users, we can contribute to the development of effective privacy protection strategies and/or user-centered design approaches. Our privacy behavior [4] may have a bigger effect than in the past as a result of greater opportunities for self-exposure on social media and the internet.

Based on the above, we argue that we need to understand how privacy behavior is formulated so as to assist policymakers to understand how users make privacy decisions and how to improve privacy behavior. There is a significant stream of work in this domain, but the academic community has provided fragmented views about what influences privacy behavior. Several researchers identify factors that determine privacy behavior relying on the lenses of a specific theory (e.g., protection motivation theory). Despite the merits of the existing scattered views, the lack of a holistic framework that aggregates those findings may prevent policy makers from taking action based on a holistic understanding of various factors that collectively affect privacy behavior. This research seeks to investigate the underlying factors that influence individuals' privacy-conscious behavior in the digital domain, as well as effective interventions to promote such behavior. Understanding how to influence human privacy-related behavior is crucial for a number of reasons. First, technological advancements cannot guarantee adequate privacy protection on their own, as the success of privacy measures ultimately depends on user behavior. Second, a significant number of privacy incidents can be attributed to individuals' lack of awareness, knowledge, or motivation to adopt privacy measures consciously.

This study can contribute to the development of strategies and interventions that may lead to enhanced privacy behavior by investigating how individuals' privacy behavior can be effectively influenced. This study examines theoretical frameworks, empirical studies, and interventions from the ICT field and other related disciplines in order to address this research gap. Our research seeks to shed light on the factors influencing privacy behavior, identify how privacy behavior is formulated, and provide actionable recommendations for ICT users, service providers, and policymakers by drawing on these interdisciplinary perspectives. To fill this gap, in this paper, we conduct an in-depth literature review to bring together these works and the factors that they identify as determinants of ICT users' privacy behavior and actions. Our research question is:

RQ: What factors have been found to influence ICT users' privacy behavior?

We argue that providing a holistic framework that aggregates the factors that influence ICT users' privacy behavior can be beneficial for many stakeholders, such as ICT users, privacy and information systems researchers, Internet Service Providers (ISP), ICT specialists, Internet Security Operation Centers (SOCs or ISOCs) (The function of the security operations center (SOC) is to monitor, prevent, detect, investigate, and respond to cyber threats around the clock. Link: https://www.checkpoint.com/cyber-hub/threat-prevention/what-is-soc/, accessed on 22 June 2023). For example, an SOC can propose and create new frameworks and services that can affect or protect privacy behavior, and thus our findings can inform SOCs towards building those frameworks and services in a way that facilitates privacy practices. As another example, researchers could contribute to the educational curriculum(s) that promote(s) an enhancement of students' privacy behavior.

In conclusion, this research provides the following main contributions:

- A list of all identified in the literature determinant factors of ICT users' privacy behavior
- A framework that synthesizes and classifies the identified factors
- A fertile environment for future research to stimulate privacy-protective behavior
- Inter-disciplinary perspective and favorable environment for further research opportunities in the privacy behavior domain

After the introduction, the paper continues with the presentation of the literature review methodology. In Section 3, we present the findings of the literature review. In Section 4, we analyze the results of the previous section. In Section 5, we discuss our results and the possible implications of our findings. Finally, Section 6 concludes the paper and provides possible solutions and future research.

## 2. Literature Review Scope and Methodology

### 2.1. Sampling Methodology

For our literature research, we used the scientific search engines of the most recognized journals in the field of information systems, also known as a basket of eight, as well as recognized international journals included in the science citation index. The "basket of eight" is considered by the Association for Information Systems (AIS) [11] one of the most representative and influential journals in the field of Information Systems. We also included the journals that are published in the proceedings of international conferences and are recognized by the AIS. We used the keywords: Privacy behavior, privacy attitude, factors that affect privacy behavior, factors that affect privacy attitude. During the initial search for factors influencing privacy behavior, studies appeared that analyze the phenomenon of "privacy paradox", so it was included as a search term in the literature review. Table 1 shows the journals that are contained in the "backet of eight" list, the journals and conferences that are associated with the AIS, and the number of articles of each journal or conference proceedings.

**Table 1.** Number of articles we chose per search source.

| Source | Number of Articles |
|---|---|
| European Journal of Information Systems (EJIS) | 55 |
| Strategic Information Systems (Search in Science Direct) | 58 |
| MIS Quarterly (MISQ) | 32 |
| International Conference on Information Systems (ICIS) | 14 |
| Information Systems Research (ISR) | 13 |
| Americas Conference on Information Systems (AMCIS) | 11 |
| European Conference on Information Systems (ECIS) | 9 |
| Journal of the Association for Information Systems (JAIS) | 12 |
| Pacific Asia Conference on Information Systems (PACIS) | 7 |
| Journal of Information Management Systems | 5 |
| Recognized international journals with citation index | 10 |
| Total number of articles | 226 |

### 2.2. Exclusion Criteria

Using the above sample methodology, we identified 945 papers. However, some of the resulting articles did not identify factors influencing privacy behavior, although they studied respective phenomena. Subsequently, we removed the studies that, although they matched the search criteria, did not focus on identifying factors that influence privacy behavior and they did not present respective results. Consequently, we excluded 648 papers because they did not specifically study our focus subject. Following this process, 297 papers remained that studied or identified specific factors that affect privacy behavior directly or indirectly. Finally, we also excluded 71 articles that studied factors that only indirectly affect privacy behavior, i.e., only factors that affect other factors that determine privacy behavior. As a result, 226 articles remained that were found to show immediate effects on privacy behavior. Figure 1 visualizes the sampling process.

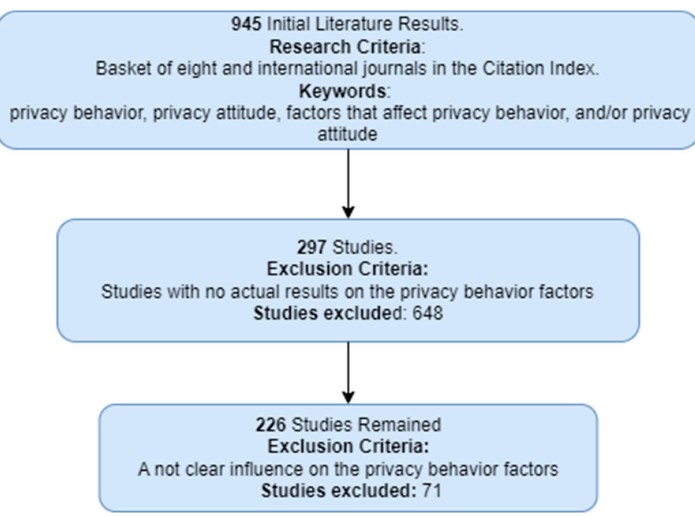

**Figure 1.** Literature sampling process and results.

Figure 2 shows the published articles by the year, and based on the results, we can notice an incremental trend per year, indicating an increased academic and research interest in the phenomenon.

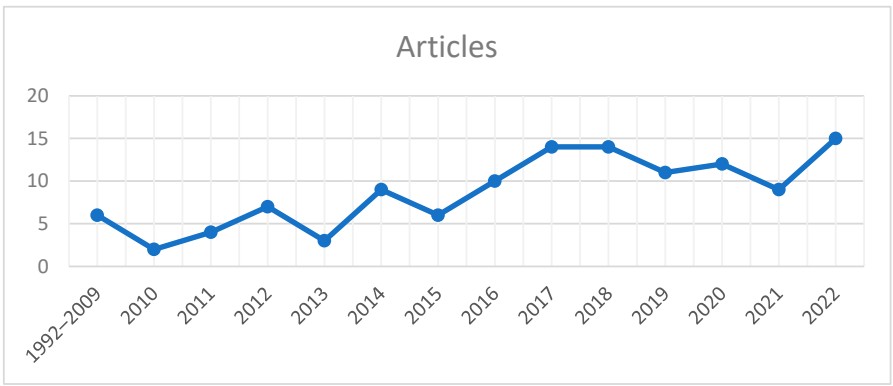

**Figure 2.** Number of articles published per year.

### 3. Factors That Affect ICT Users' Privacy Behavior

We studied each one of the papers selected using the methodology analyzed in Section 2, seeking the identification of factors that influence the privacy behavior of ICT users. Following this process, we identified eleven dominant factors. In studies that generally examine the factors that influence privacy behavior, the most prevailing factors are privacy concerns and the perception of risk. From the analysis of the current literature, the factors that most often appear in the literature to influence ICT users' privacy behavior are (Table 2). We have to mention that the studies were conducted with different criteria and methods. Some were conducted with qualitative methods and others with quantitative methods and different variables as input. We also have to mention that there is no scientific consensus on the definition of the term privacy behavior, and it depends on the view or the objective of the study.

In some cases, we categorized the above factors into broader clusters depending on how much they have in common. For example, gender, age, income, and political position have been categorized in the group "Demographics" as a single cluster. Similarly, the necessity, needs, and psychological entanglement formed a cluster. To make it clearer, as a necessity we define the need for someone to provide personal data to an authority for a specific reason, such as to board a plane or to book a hotel room. As "needs" or psychological entanglement, we define the need of someone to receive a non-mandatory product or service and they must provide their personal data in order for this to happen.

We have to define that we do not depict any relationships between these two factors. The remaining clusters are: Financial exchanges, financial benefits, and usefulness as one cluster, trust, control, confidence, and fear as another cluster, education, visualization, interaction and experience as another cluster, and finally, the dimensionality and the complexity of taking a privacy decision as a cluster. Following this clustering process, we created the following conceptual scheme shown in Figure 3, while Table 3 aggregates the research (checked in the table) that studied what influences ICT users' privacy behavior.

**Table 2.** Number of Articles per Factor.

| Factors | Number of Articles That Identify the Factor |
|---|---|
| Financial Exchanges/benefits/usefulness | 31 articles |
| Privacy Risk Perception | 27 articles |
| Trust/Control/confidence/fear | 35 articles |
| Privacy Concerns | 63 articles |
| "Needs" psychological engagement/necessity | 16 articles |
| Sensitivity of information | 3 articles |
| Privacy Awareness | 21 articles |
| Time lapse | 7 articles |
| Education/Visualization/Interaction/ Experience | 19 articles |
| Demographics (age/gender/country, political position, income, etc.) | 24 articles |
| Dimensionality/Complexity of a privacy decision making | 10 articles |

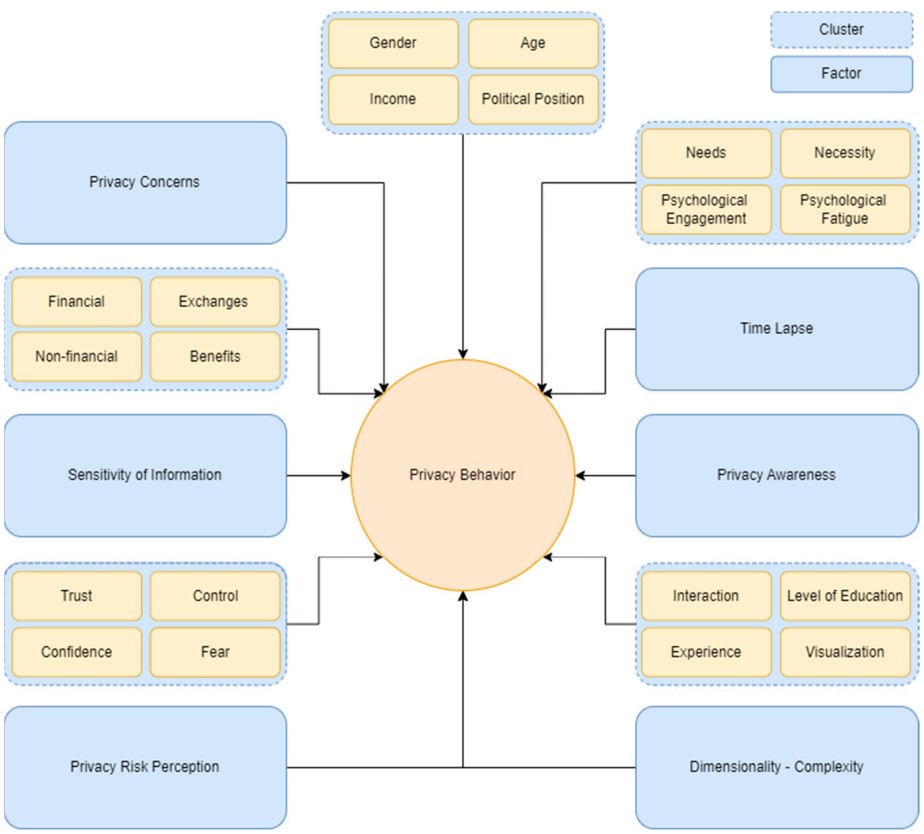

**Figure 3.** Conceptual scheme.

**Table 3.** Factors influencing ICT users' privacy behavior in the literature.

| References | Demographics | Privacy Risk Perception | Financial Exchanges/ Benefits/Fatigue | Needs Necessity | Privacy Concerns | Trust/Control/ Confidence/Fear | Sensitivity of Information | Time | Education/Visualization/ Interaction/Experience | Dimensionality/ Complexity | Privacy Awareness |
|---|---|---|---|---|---|---|---|---|---|---|---|
| [12] Acquisti et al. (2015) | | | | | √ | | | | | | |
| [13] Alashoor et al. (2018) | | | | √ | | | | | | | |
| [14] Alawadhia and Hussain (2019) | | | | | √ | | | | | | |
| [15] Ameen et al. (2021) | | | | | √ | √ | | | | | |
| [16] Arica et al. (2022) | | | | | √ | | | | | | |
| [17] Avshalom and Yaron (2017) | √ | √ | √ | | √ | √ | | | √ | | |
| [18] Ayaburi and Treku (2020) | | | | | √ | √ | | | | | |
| [19] Bal (2014) | | √ | | | | √ | | | | | |
| [20] Bachura et al. (2022) | | | | | | √ | | | | | |
| [21] Becker (2018) | | | | | √ | | | | | | |
| [22] Bhagat et al. (2018) | | | | | | | | | √ | | |
| [23] Buchanan et al. (2007) | | | | | | | | | | √ | |
| [24] Buck (2017) | | | | | √ | | | √ | | | |
| [25] Cerruto et al. (2022) | | | | | √ | | | | | | |
| [26] Chakraborty et al. (2013) | √ | | | √ | | | | | | | |
| [27] Chawla and Kumar (2021) | | | | | | √ | | | | | |
| [28] Choi et al. (2018) | | | √ | √ | √ | | | | | | |

**Table 3.** *Cont.*

| References | Demographics | Privacy Risk Perception | Financial Exchanges/Benefits/Fatigue | Needs Necessity | Privacy Concerns | Trust/Control/Confidence/Fear | Sensitivity of Information | Time | Education/Visualization/Interaction/Experience | Dimensionality/Complexity | Privacy Awareness |
|---|---|---|---|---|---|---|---|---|---|---|---|
| [29] Chou et al. (2019) | √ | √ | | | | | | | | √ | |
| [30] Cloarec et al. (2022) | | √ | √ | | √ | √ | | | | | |
| [31] D'Souza and Phelps (2009) | | | √ | | √ | | | | | | |
| [32] Davazdahemami et al. (2018) | | | | | √ | √ | | | | | |
| [5] Dhir et al. (2016) | √ | | | | | | | | | | |
| [33] Dienlin and Trepte (2015) | | | | | √ | | | | | √ | |
| [34] Ermakova et al. (2014) | | | √ | | √ | √ | | | √ | | |
| [35] Figl et al. (2020) | | | | | | | | | √ | | |
| [36] Flender and Müller (2012) | | | | | | | | √ | | | |
| [37] Fox et al. (2018) | | | | | √ | | | | √ | | √ |
| [38] Gabel et al. (2019) | | | √ | | √ | | | | | | |
| [39] Gaurav (2008) | | | | √ | √ | | | | | | √ |
| [4] Gerber et al. (2018) | √ | √ | | | | √ | | √ | √ | | √ |
| [40] Ghose et al. (2020) | | √ | | | √ | √ | | | | | |
| [41] Gómez-Barroso (2018) | | | √ | | √ | | | | | | |
| [2] Hallam and Zanella (2017) | | | √ | | √ | | | | √ | | |
| [42] Hatamian et al. (2019) | | √ | | | | | | | | | |

**Table 3.** *Cont.*

| References | Demographics | Privacy Risk Perception | Financial Exchanges/ Benefits/Fatigue | Needs Necessity | Privacy Concerns | Trust/Control/ Confidence/Fear | Sensitivity of Information | Time | Education/Visualization/ Interaction/Experience | Dimensionality/ Complexity | Privacy Awareness |
|---|---|---|---|---|---|---|---|---|---|---|---|
| [43] Heravi et al. (2018) | √ | | | √ | √ | | | | √ | √ | √ |
| [44] Hew et al. (2017) | √ | | √ | | | | | | | | |
| [45] Hofstra et al. (2016) | √ | | | | √ | √ | | | | | |
| [46] Ioannou et al. (2020) | | | √ | √ | √ | √ | √ | | √ | | |
| [47] Ioannou and Tussydiah (2021) | | | | | √ | √ | | | | | |
| [48] Jensen et al. (2017) | | √ | | | √ | | | | | | |
| [49] Jeong and Kim (2017) | √ | | | | √ | | | | | | |
| [50] Jia and Xu (2016) | | | | √ | √ | | | | | √ | |
| [51] Jiang (2018) | √ | | | | | | | | | | |
| [52] Johnson (2013) | | | | | | | | | √ | | |
| [53] Jordaan and Van Heerden (2017) | | | | √ | √ | √ | | | | | √ |
| [54] Jozani et al. (2020) | | √ | √ | | | | | | | | |
| [55] Junga and Park (2018) | | | | | √ | | | | | | |
| [56] Kang et al. (2016) | | | | √ | | √ | | | | | |
| [57] Kayes and Iamnitchi (2017) | | | | | | | | | | | √ |
| [58] Keith et al. (2012) | | | √ | | | √ | | | | | √ |
| [59] Keith et al. (2014a) | | √ | √ | | | | | √ | | | |

**Table 3.** *Cont.*

| References | Demographics | Privacy Risk Perception | Financial Exchanges/Benefits/Fatigue | Needs Necessity | Privacy Concerns | Trust/Control/Confidence/Fear | Sensitivity of Information | Time | Education/Visualization/Interaction/Experience | Dimensionality/Complexity | Privacy Awareness |
|---|---|---|---|---|---|---|---|---|---|---|---|
| [60] Keith et al. (2014b) | | | √ | | √ | | | | | | |
| [61] Kim et al. (2022) | | | | | √ | | | | √ | | |
| [62] Kitsios et al. (2022) | | | | | | √ | | | | | |
| [63] Knijnenburg et al. (2013) | √ | | | | √ | | | | | √ | √ |
| [64] Korunovska et al. (2020) | | | | √ | | | | | √ | | |
| [65] Kosinski et al. (2013) | | | | | | | | | √ | | |
| [66] Krasnova et al. (2014) | | | √ | | | | | | | | |
| [67] Kraus et al. (2017) | | | | √ | | | | | | | |
| [68] Kurt (2010) | √ | | | | √ | | | | √ | | |
| [69,70] Kwee-Meier et al. (2016a,b) | | √ | | | √ | | | | | | |
| [71] Lankton et al. (2017) | √ | √ | √ | | √ | √ | | | √ | | √ |
| [72] Lee et al. (2022) | | | √ | √ | | | | | | | |
| [73] Li and Chau (2019) | | | | | √ | √ | | | | | |
| [74] Li et al. (2015) | √ | √ | | | √ | | √ | | | | |
| [3] Li et al. (2017) | √ | | | | √ | √ | | | | √ | |
| [75] Li et al. (2019) | | | | √ | | | | | | | |
| [76] Li et al. (2020) | √ | √ | √ | | | | | | | | |

**Table 3.** *Cont.*

| References | Demographics | Privacy Risk Perception | Financial Exchanges/ Benefits/Fatigue | Needs Necessity | Privacy Concerns | Trust/Control/ Confidence/Fear | Sensitivity of Information | Time | Education/Visualization/ Interaction/Experience | Dimensionality/ Complexity | Privacy Awareness |
|---|---|---|---|---|---|---|---|---|---|---|---|
| [77] Li et al. (2022) | | | | | √ | | | | | | |
| [78] Liao et al. (2011) | | | | | √ | | | | | | |
| [79] Lidynia et al. (2018) | | | | | √ | | | | | | |
| [80] Lu et al. (2020) | | √ | | | √ | | | | | | |
| [81] Mager et al. (2021) | | | √ | | | | | | | | |
| [82] Marreiros et al. (2017) | | | | | | | | | √ | | √ |
| [8] Mathews-Hunt (2016) | | | | | √ | √ | | | | | √ |
| [83] McCoy et al. (2017) | | | | | | | | √ | | | |
| [7] Menard and Bott (2020) | | √ | | | √ | √ | | | | √ | |
| [84] Mosafer et al., 2021 | | √ | √ | | √ | | | | | | |
| [85] Moshki and Barki (2014) | | | | | √ | | | | | √ | |
| [86] Mousavi et al. (2022) | | | | | √ | | | | | | |
| [87] Mullins et al. (2022) | | | | | √ | | | | | | |
| [88] Mutimukwe et al. (2020) | | √ | | | √ | √ | | | | | |
| [89] Nikkhah and Grover (2022) | | | | | | √ | | √ | | | |
| [90] Nikkhah and Sabherwal (2017) | | | √ | | √ | √ | | | | | |
| [91] Niknejad et al. (2020) | | | | √ | √ | √ | | | | | |

**Table 3.** *Cont.*

| References | Demographics | Privacy Risk Perception | Financial Exchanges/ Benefits/Fatigue | Needs Necessity | Privacy Concerns | Trust/Control/ Confidence/Fear | Sensitivity of Information | Time | Education/Visualization/ Interaction/Experience | Dimensionality/ Complexity | Privacy Awareness |
|---|---|---|---|---|---|---|---|---|---|---|---|
| [92] Nyshadham and Castano (2012) | | √ | | | | √ | | | | | |
| [9] Palos-Sanchez et al. (2019) | | | | | √ | | | √ | | | |
| [93] Park (2011) | √ | | | | | | | | | | |
| [94] Park (2015) | √ | | | | | | | | | | |
| [95] Paspatis et al. (2020) | | | | | | | | | | √ | √ |
| [96] Pilton et al. (2021) | | | | | | | | | | | √ |
| [97] Quayyum et al. (2021) | √ | √ | | | | | | | | | √ |
| [98] Rangedda et al. (2022) | | | √ | | | | | | | | |
| [99] Reith et al. (2019) | | | √ | | √ | √ | | | | | |
| [100] Reith et al. (2021) | | | | | √ | √ | | | | | |
| [101] Renaud and Zimmermann (2018) | | | | | | | | | | √ | √ |
| [102] Reynolds et al. (2011) | √ | | | | | | | | | | √ |
| [103] Risius et al. (2020) | | | | | | √ | | | | | √ |
| [104] Schomakers et al. (2019) | √ | √ | | | | | √ | | | √ | |
| [105] Schreiber et al. (2013) | | | | √ | √ | √ | | | | | |
| [106] Schreiner and Hess (2015) | | √ | √ | √ | | √ | | | | | |
| [107] Segura et al. (2018) | | | | | √ | | | | | | |

Table 3. *Cont.*

| References | Demographics | Privacy Risk Perception | Financial Exchanges/Benefits/Fatigue | Needs Necessity | Privacy Concerns | Trust/Control/Confidence/Fear | Sensitivity of Information | Time | Education/Visualization/Interaction/Experience | Dimensionality/Complexity | Privacy Awareness |
|---|---|---|---|---|---|---|---|---|---|---|---|
| [108] Senarath and Arachchilage (2018) | | √ | | √ | √ | √ | | | | | |
| [6] Shane-Simpson et al. (2018) | √ | | | | √ | | | | | | |
| [109] Sharma and Crossler (2014) | | √ | √ | | √ | | | | | | √ |
| [110] Spiekermann et al. (2012) | | | √ | | √ | | | | | | |
| [111] Sschwaig et al. (2013) | | | | | √ | | | | | | |
| [112] Strycharz et al. (2021) | | | | | | √ | | | | | |
| [113] Stutzman et al. (2011) | √ | | | | √ | | | | | | |
| [114] Taddicken (2014) | | | | | √ | | | | | | |
| [115] Terlizzi et al. (2019) | | | | √ | √ | √ | | | | | |
| [116] Tsai et al. (2011) | | | | | √ | √ | | | | | |
| [117] Tsai and Kelley (2014) | | | | | √ | | | | √ | | |
| [118] Tse et al. (2014) | | | | | | | | | | | √ |
| [119] van Zoonen (2016) | | √ | | | √ | √ | | | | √ | √ |
| [120] Venkatesh et al. (2012) | | | √ | | | | | | | | |
| [121] Viswanath et al. (2020) | | √ | √ | | | √ | | | | | |
| [122] Wall and Warkentin (2019) | | | | | √ | √ | | | | | |
| [123] Wang et al. (2021) | | | | | | √ | | | | | |

**Table 3.** *Cont.*

| References | Demographics | Privacy Risk Perception | Financial Exchanges/ Benefits/Fatigue | Needs Necessity | Privacy Concerns | Trust/Control/ Confidence/Fear | Sensitivity of Information | Time | Education/Visualization/ Interaction/Experience | Dimensionality/ Complexity | Privacy Awareness |
|---|---|---|---|---|---|---|---|---|---|---|---|
| [124] Wiegard and Breitner (2017) | | | √ | | | | | | | | |
| [125] Wieneke et al. (2016) | | √ | | | | | | | | | |
| [126] Wilson and Valacich (2012) | | √ | √ | | | | | | | | |
| [127] Wilson et al. (2015) | | | | | √ | √ | | | | | |
| [128] Wisniewski et al. (2017) | | √ | | | | | | | | | √ |
| [129] Wu and Li (2019) | | | √ | | | | | | | | |
| [130] Xu et al. (2010) | | √ | √ | | √ | √ | | | | | |
| [131] Zhang et al. (2020) | | | | | √ | √ | | √ | | | |
| [132] Zareef and Gurvirender (2015) | | | | | | | | | | √ | |
| [133] Zalmanson et al. (2022) | | | | | √ | √ | | | | | |
| Number of references per factor | 23 | 30 | 34 | 16 | 74 | 45 | 3 | 8 | 14 | 8 | 22 |

### 4. How Behavioral Factors Affect ICT Users' Privacy Behavior

Depending on the nature of each factor that affects privacy behavior, there is a different perspective on how and why it affects privacy behavior, while almost all factors get affected and affect the above behavior positively and/or negatively. Factors influencing privacy behavior can be differentiated according to their effect (i.e., how strongly they tend to influence behavior according to the relevant study) or depending on how often they appear as an influencing factor in the literature or the type of influence they exercise on behavior (i.e., positive, negative). There is no existing classification of privacy behavior determinant factors in the literature. In this work, we bring together the various factors that had been identified by researchers. Those factors are driven by multiple theories (e.g., protection motivation theory) or the researchers' point of view. Thus, they vary in level of abstraction, nature, and scope. Creating classification categories and assigning the identified factors into the relevant categories was a challenging task to ensure consistency and avoid duplication of factors in more than one category. For example, we examined the factors needs: When is something necessary, "needs": When is something psychologically necessary, and necessity under the same cluster. However, even though some factors, such as privacy risk perception, privacy concerns, and privacy awareness, have similar meanings or outcomes, we chose to separate them because many researchers have examined them separately and their findings present different results. For instance, the phenomenon of the privacy paradox is more associated with privacy concerns than with privacy awareness and privacy risk perception [1–4], while privacy awareness examines a more general behavior. On the other hand, privacy risk perception is highly associated with the potential result of someone's privacy action. Among the 11 factors we identified in the literature, we found in the respective papers that six factors show strong influence (i.e., quantitative studies have verified a strong correlation between them and privacy behavior). From the 11 factors, the literature analysis showed that eight of them appear frequently as determinants of privacy behavior. From the 11 factors, four show only positive influence on privacy behavior, two only negative behavior, and the remaining five factors may have positive or negative behavior depending on the situation of appearance.

#### 4.1. Demographics

We discuss the factors "gender and age" of the "Demographics" cluster. Research shows that gender plays an important role in the self-exposure of personal information on the internet [26,45,51,94,102], with age contributing to it [4,26,45,71,73]. For example, females tend to post more often and with fewer privacy restrictions on OSNs than males, while older users claimed to be more concerned with privacy, and this is reflected in their posting practices [107]. However, it is important to note that there are contrasting findings in the literature as well. One study found that demographics, such as gender, may have little to no influence on privacy behavior [68]. These contradictions suggest that the relationship between gender and privacy behavior is complex and may be influenced by various contextual factors. Because this study was conducted in an Asian country (Turkey), in contrast to all others, we assume that ethnicity or even religion or a combination of them may play a significant role in privacy behavior. The combination of gender and age plays a role in choosing which OSN will be preferred for a post, with female students preferring Facebook and male students preferring Instagram [6]. Adolescent men tend to post less data on OSNs and untag their older photographs as they age [5]. Even political position plays a role in US citizens, with the Democrats being more privacy-concerned than Republicans [40]. In the same research, authors found that low-income populations and females were more privacy-conscientious and more likely to opt out of location-tracking apps.

In conclusion, while there are contradictions in the literature regarding the influence of gender and other demographic factors on privacy behavior, there is a consensus that gender and age can indeed shape individuals' self-exposure of personal information on the internet. The complex nature of these relationships calls for further research to explore the

underlying mechanisms and contextual factors that contribute to privacy behaviors across different demographic groups.

### 4.2. Privacy Risk Perception

The perception of risk directly affects the privacy behavior of a user when (s)he wants to post personal information on the internet [80] and especially to OSNs [5,6]). In the e-commerce context, consumers tend to undervalue the probability of risks and have difficulty separating their existing risk exposure from potential new threats [58]. The higher the risk perception, the lower the information that a user shares on the internet [58,84] and limiting the posts to OSNs [5]. Nonetheless, even if risk perception affects privacy behavior, this is mediated by financial exchanges/benefits. Further, privacy risk perception seems to have a stronger influence on privacy behavior when the person is young, and as the individuals become more mature, their privacy behavior is modified [29]. In combination with other factors from the cluster demographics, such as age [4,26,45,71,73] and gender [102], individuals weigh the risk of self-disclosure against the financial exchanges/benefits and modify their privacy behavior accordingly [84,109]. If the individual considers that the risk of disclosure is greater than what they can accept in relation to the perceived benefit, then they will disclose less personal information [19,84,109] or they will reject the financial offer. Reith et al. [99] research showed that ICT users' risk perception is important when they must choose what kind of mobile payment solution, they should choose to make a payment on the internet [99]. Terlizzi et al. [115] research showed similar results. More specifically, their research showed that privacy risk perceptions were also important when ICT users had to choose and use a banking application, and users tend to prefer the applications they trust more. In conclusion, while there are some inconsistencies in the literature, the impact of risk perception on privacy behavior is clear. Individuals' decisions about self-disclosure and information sharing are influenced by the interplay of risk perception, financial exchanges/benefits, demographics, and trust. As we mentioned in Section 4.1, more research is required to delve deeper into these relationships and comprehend the nuanced factors that shape privacy behavior in various contexts.

### 4.3. Financial Exchanges, Benefits, and Fatigue

Financial and non-financial exchanges and benefits are also factors that have a strong and direct effect on privacy behavior. According to the literature, individuals seem receptive to modifying their behavior according to the economic and non-economic exchanges, regardless of the beliefs they state to hold and the privacy awareness and privacy concerns they state to have [46,54,71,81,90,109,130]. Financial and non-financial exchanges, even if they do not change the individuals' perception, are able to persuade individuals to put aside their fears and give their personal data to third parties and consent to the terms they will be asked to accept [41,44,54,127,130]. Experimental evidence provides insights into individuals' behavior regarding financial and non-financial exchanges. For instance, in an experiment in 2021 with 1274 consumers, the majority consented to trade their internet cookies for a 10-euro coupon value [81]. In another experiment, Schreiner and Hess [106] found that consumers are willing to pay an overhead amount of money for a privacy-freemium model in a service they want to be provided. Krasnova et al. [66] research also found that users are willing to change their privacy protection settings on mobile apps in exchange for a lesser price or extended functionality, with the functionality being less valued than the cost.

Healthcare patients seem to accept disclosing personal health information if the output benefit compensates for the value of the personal information [38]. Additionally, experimental evidence suggests that the limitation of the information that is given, rather than a deliberate evaluation of costs and benefits of privacy, affects people's privacy behaviors [92]. Alashoor et al. [13] showed that the positive effect of perceived benefits on disclosure likelihood was amplified under a positive mood state, whereas the negative effect of perceived privacy risks on disclosure likelihood was trivial under a positive mood state.

Some researchers also cite privacy fatigue as a category of non-financial rewards [28,59], with consumers stating that the fatigue comes from the privacy control complexity and that it affects their self-disclosure behavior [59]. Choi et al. [28] conducted a survey with 324 internet users to examine how privacy fatigue affects privacy behavior. Their findings showed that privacy fatigue (or, according to them, privacy burnout) has a stronger impact on privacy behavior than privacy concerns. This factor showed a strong correlation with other factors, such as privacy perception and privacy concerns [84,99,100]. In another example, Reith et al. [99] conducted research with 466 participants that got paid with a 20-euro voucher. Their results showed that gadget lovers would step aside from their privacy concerns while using mobile payment solutions [99]. Finally, the literature highlights the importance of financial and non-financial exchanges, privacy fatigue, and the importance of information in shaping privacy behavior. While there are some contradictions in the literature, it is clear that individuals are influenced by a variety of factors when making decisions about self-disclosure. The complex interplay of these factors necessitates additional research to better understand the mechanisms underlying privacy behavior in various contexts.

### 4.4. Needs and Necessity

Another important factor found to influence privacy behavior is the necessity of the information for the service provided. This factor seems to influence whether disclosure of information is required for the satisfaction of a real need, such as to present your identification documents to an aviation company when traveling [28,46], and in some cases, maybe even a medical necessity enforces to show medical records, such as COVID vaccination certificate [134] or a virtual need e.g., the need that has arisen from psychological charging [28,53,67] such as downloading an application from the internet that requires personal data for its download [45] or the need that arises for socialization in ONSs [26,53].

### 4.5. Privacy Concerns

Another factor influencing privacy behavior is privacy concerns. The factor privacy concerns seem to affect other clusters of factors [28,46,74] and is affected [49,73] by almost all the factors mentioned above. For example, it seems to influence and be influenced by the trust factor that an individual shows [46,47,90,127] and is influenced by the lack of awareness and risk perception factor [37,53,73,119]. Nikkhah and Sabherwal [90] found that the main inhibitor of disclosing personal information to Mobile Cloud-Computing apps (MCC) is perceived privacy concerns, and the main enablers are perceived usefulness and trust. It is worth noting that this factor has been discussed at length in relation to the paradox of privacy. According to studies, privacy concerns, although reported by many study subjects, do not seem to have a real effect on privacy behavior [1–4,103]. They may partially alleviate the attitude of users to post personal information on the internet, but eventually, users do adopt this behavior [80]. In conclusion, the literature highlights privacy concerns as an influential factor in privacy behavior, with intricate relationships with other factors. However, contradictions arise regarding the actual impact of privacy concerns on individuals' behavioral choices. Despite reporting concerns, individuals often proceed with posting personal information online. These contradictions underscore the complexity of the relationship between privacy concerns and behavior, necessitating further research to gain a deeper understanding of the underlying mechanisms and dynamics.

### 4.6. Trust, Control, and Confidence

Trust in an application or a service affects an individual's privacy behavior, and when it is acquired, it bypasses other factors, especially in the presence of the factor age of the cluster demographics (i.e., when users are older, trust acts more as a factor that bypasses other factors) [3]. According to Xu et al. [130], trust could play a primary role in addressing privacy concerns pertaining to OSNs, especially in the absence of well-

established legal resources. Other models state that trust reduces privacy concerns [90,105] and thus affects privacy behavior. If there are communication channels that effectively build trust or be transparent to consumers about how their personal data are processed while using mobile apps consumers' privacy behavior can be modified [47,73,90,105]. For example, Wilson et al., [127] proposed a theoretical model and validated it through a controlled experiment that shows how empowering individuals with a sense of control over their personal information can help mitigate their privacy concerns. According to Ermakova et al. [34], trust plays a significant role in mitigating privacy concerns when a patient needs to upload medical records to a cloud service. Another study suggests that transparency of the practices of service leads to more accurate risk and trust perceptions and provides an improved foundation for informed decision-making [34]. In another study, the findings show that the important roles of political trust and the belief that governments need to be proactive in protecting peoples' welfare during a crisis can increase acceptance of surveillance and thus assist in the management of the health crisis [47]. Even though the literature contains contradictions, the influence of trust on privacy behavior is evident. Trust can act as a significant mitigating factor for privacy concerns and influence individuals' personal information-related decisions [1–4]. The complex and multifaceted character of trust, however, necessitates additional research into its dynamics and effects in various contexts.

### 4.7. Education, Interaction, Experience, Sensitivity of Information, Visualization, and Time-Lapse

Other factors found, such as: (a) The level of education, (b) the time available to make a decision [60], and (c) the sensitivity of the information we may need to disclose [74], showed having little influence on privacy behavior. A person's level of education affects almost all the above factors [29,104]. Some studies show even a strong influence of education on privacy behavior depending on the presence of other factors, such as privacy concerns [43]. In addition, privacy-related knowledge, privacy-related IT knowledge, and general involvement with IT-related privacy seem to reduce privacy concerns [85].

The time available to make a decision to give personal data seems to affect our privacy behavior [60,101]. Studies show that the less response time is available for the individual to decide to give his/her personal information, the more expected a person's behavior is depending on the current level of privacy awareness. Thus, people with greater patience and self-control are less willing to disclose information through apps when first presented with the choice [59]. In addition, individuals are less tempted to disclose information as the time frame of risks is extended further into the future. The available time between a given information and making a decision in a complex matter seems to influence the decision in a potentially negative way. When an event lasts or is repeated at frequent intervals, it seems to affect privacy behavior [83,101].

In combination with the above paragraph, when a frequent interval event uses visual stimuli, such as repetitive advertisements, they seem to modify privacy concerns [9]. As soon as people get the knowledge and acquire the experience that someone is interested in their personal information, they value their information much more than before [110]. Thus, these people may alter their privacy behavior to a more secure one. In addition, privacy behavior can be improved by using visualization technics, such as privacy labels that promote privacy best practices to consumers [37] or better privacy policy designs [60]. In a relationship with the above paragraph and the factor time-lapse, Buck's [24] results on how time and experience affect privacy behavior show a significant effect on the concerns users have about their privacy—an increasing future self-continuity is related to greater concerns. Regarding visualization, according to Fox et al. [53], privacy labels seem to enhance privacy knowledge and reduce consumers' privacy concerns. In addition, Figl et al. [35] concluded that privacy nudging encourages users to choose the settings that meet their privacy needs without increasing any cognitive cost.

*4.8. Privacy Awareness*

Finally, one factor that can have a small to a very large effect depending on the context is how privacy-aware a person is [39,43,74]. This factor influences and is influenced by other factors, such as risk perception, and is strongly associated with privacy concerns and demographic factors (age, gender) [4,26,45,71,74], and level of education [5,74]. Furthermore, a person's lack of privacy awareness can heighten privacy concerns. Tse [118] stated that only a few persons change the default privacy protection configuration in OSNs, due to a lack of basic security knowledge and the perspective of perceptible threats. Risius et al. [103] found that if users' privacy education is successful in building short-term commitment and intentions, that translates into actual privacy-protecting behaviors. Pilton et al. [96], during their research, created a web-browser extension called "paradox" that modifies a website's privacy policy to an easier-to-read one. Their results showed that users' privacy awareness was enchanted to a more secure one. The same results showed in Paspatis et al. [95] research. Their browser extension for mobile applications called "AppAware extension" and AppAware Client-Server App was modifying apps' privacy policy using the app's permission set to an easiest-to-read one. After a questionnaire survey, ICT users' who participated in the survey stated that their privacy awareness was enhanced after using the AppAware App.

**5. Discussion**

Our research has shed light on the various factors that have been found to influence ICT users' privacy behavior. Various research works show that different factors influence privacy behavior and, thus, it is a multi-factor issue. From the systematic literature review that we performed, we aggregated the multiple factors that affect individuals' privacy behavior. We organized the identified factors into groups of eleven clusters. Our classification also includes per factor the severity of influence on privacy behavior, the frequency in which it is mentioned by researchers, and whether the factor has a positive or negative effect (or both).

The identified factors seem to influence privacy behavior with different strengths depending on the context in which it takes place, as well as the subject that it is influenced. Not all researchers have agreed on the strength and significance of the influence that each factor has. One possible explanation for these variations could be that the different empirical works were conducted in different cultures or with different sample sizes. For example, Kurt [68] conducted a survey at the University of Ankara with $n = 163$ females and $n = 42$ male students participating in privacy behavior and OSNs. The researcher found out that "there is not any statistical significance between privacy behaviors and genders" when it comes to posting personal information on OSNs. Kurt's results [68] are opposite to the majority of other research works that point out that gender plays a significant role in the self-exposure of personal information on the internet [26,45,51,74,94,102]. However, he does not offer any explanation regarding this deviation. One potential explanation could be the cultural characteristics of the sample, given that the participants were only students and only from one country.

Another finding from our research refers to the difficulty of studying existing literature and compare existing works due to the lack of common terminology in the academic community that studies the subject. The respective researchers utilize different terms to describe the same concept or interpret in different ways the same term. For example, researchers Nikkhah and Sabherwal [90] use the risk factor when they actually refer to the trust factor. In another example, researchers Ermakova et al. [34] discuss the trust factor when they seem to actually refer to the concept of control. This is also evident when encountering research on the time-lapse factor. Some researchers use the time factor, while they actually discuss the age of the individual, e.g., people tend to publish more personal data to the OSNs when they are young [5].

Based on the analysis of existing works, we have clustered the identified factors and we presented a collective view of the factors that influence individuals' privacy behavior.

Within the scope of our research, we noticed that some of the factors of privacy behavior appear in the literature with particular frequency. The fact that several studies have addressed the "privacy paradox" phenomenon has greatly influenced the privacy concerns factor, with 74 surveys analyzing, discovering, or studying it. Demographics have been analyzed by 23 surveys. As this factor contains sub-factors, such as age, gender, etc., it creates favorable conditions for its appearance. Several studies dealing with privacy behavior were analyzed by groups such as gender, age groups such as adolescents, students, adults, or combinations thereof. The third most common factor is the perception of privacy risk. This can be explained by the fact that this factor seems to indirectly affect almost all other factors and directly the privacy behavior but it is also influenced by other factors.

*5.1. Recommendations for the Private Sector and Practitioners*

This paper provides findings from literature analysis regarding what influences ICT users' privacy behavior, which can be beneficial to the private sector. Companies and ICT or IT individual specialists that create mobile applications or application stores can utilize our findings to modify the way that they present to the users their privacy policies, i.e., what personal data they collect, why they need them and store them, and how they use them or to show that they satisfy the data minimization principle. Our analysis showed two things, respectively. As our analysis demonstrates, the factor of financial exchanges and benefits affects privacy behavior, and it becomes evident that ICT users are willing to pay more for a premium application that promotes personal data transparency or even personal data-free collection—privacy premium [106]. Respectively, from the cluster trust, control, and confidence, it becomes evident that ICT users tend to choose mobile applications and online stores if they feel secure about their personal data or if they feel they have control over them [73,90,105]. Companies that offer SoC as a service (SoCaaS) can also benefit from our research, for example, by utilizing the demonstrated privacy behavior factors they can create and apply privacy policies for their clients and at the same time, they can identify the weak spots in a company and recommend applying more privacy and security controls. ISPs can suggest and offer new privacy controls to their clients by knowing what affects privacy behavior, i.e., more parental control when they know that minors and adolescents are less inclined to apply protective privacy behaviors when using the home internet for browsing, school homework, and using their social media.

*5.2. Recommendations for Policy Makers and Educational Institutes*

The understanding of what affects privacy behavior can create a favorable ground in academia for further research. Teachers and researchers can use the knowledge that results from understanding the factors that influence privacy behavior to create teachable and learned knowledge. For example, professors and teachers can utilize our findings to create specialized information privacy courses or theoretical/laboratory modules. The findings of this paper can also be used by designers of privacy awareness programs to promote privacy behavior. Our literature review showed that younger people tend to publish often personal data, and this is probably due to the lower risk perception they have. Thus, the state and respective authorities who design privacy awareness campaigns can benefit by targeting young people and students aiming to augment their privacy awareness and raise their risk perception levels. In our opinion, our review showed that what affects privacy behavior remains an open research subject. Focusing and trying to exploit the factors we analyzed, we may find new research paths and try to understand why the aforementioned factors influence privacy behavior.

## 6. Conclusions and Future Work

Privacy behavior seems to have an important role in our daily computerized life. More than two hundred studies researched and tried to identify the reasons behind our privacy actions while we browse the internet. They proposed various reasons and solutions, but still, we are not sure if we can see the whole picture behind what influences ICT users' privacy

behavior. The studies we reviewed in this paper revealed multiple factors that, under the proper conditions, can influence privacy behavior. To the best of our knowledge, this is the first literature review that studied the set of factors that influence privacy behavior and not an individual factor or individual. We argue that our findings provide an answer to the research query posed in the introduction of what factors can influence ICT users' privacy behavior. To the best of our knowledge, this is the first research that concentrated on privacy behavior determinant factors, examined them, and categorized them into single factors and clusters of factors. We believe that our results can shift the attention of the academia and private companies to more targeted directions to provide better services and products to ICT users, as we mentioned in the previous section. We will continue our research by studying how we can exploit the knowledge of the identified factors to alter ICT users' privacy behavior towards more protective privacy actions. We aim to study how those factors were utilized in other fields, such as psychology, sociology, and philosophy, in order to modify behaviors. Our target is to be able to empirically explore through experiments how the technological context can trigger the modification of privacy behaviors and understand how privacy behavior is formulated in real technological contexts (e.g., e-government services, mobile applications), and propose mechanisms to promote protective behaviors. Our future research will focus on utilizing the aforementioned privacy behavior determinant factors to propose techniques to enhance protective privacy behavior, based on similar research that had been conducted in other fields, such as psychology, sociology, sociology, and health.

Aside from the above, our investigation reveals numerous research opportunities. Changes in privacy behavior are a subject that still requires extensive study. Researchers may examine the efficacy of psychological factors in privacy behavior change and the durability of behavior modification interventions over the long term. Our research may create opportunities for further research in several areas, such as the utilization of privacy determinant factors to enable privacy-protective behaviors.

Our findings provide a comprehensive perspective on the significance of privacy behavior enhancement and inspire further research by addressing the potential benefits and research opportunities and promoting inter-disciplinary collaborations.

This limits our study because it leads to the lack of a definition of what constitutes a strong or weak influence that a specific privacy behavior factor has on privacy behavior. Therefore, our findings related to the influence of privacy behavior determinant factors (i.e., Table 4) are limited, respectively.

**Table 4.** Severity, Frequency, and Influence of Factors.

| Factor | Strong Influence | Frequency of Appearance in Literature | Positive Influence | Negative Influence |
|---|---|---|---|---|
| Financial/Non-Financial Exchanges/Benefits/Usefulness | √ | √ | √ | √ |
| Privacy Risk Perception | √ | √ | √ | √ |
| Trust/Control/Confidence/Fear | | √ | √ | |
| Privacy Concerns | √ | √ | √ | |
| Needs/necessity/Psychological Engagement | | √ | | √ |
| Sensitivity of information | | | √ | |
| Privacy Awareness | √ | √ | √ | √ |
| Time Lapse | | | √ | |
| Level of Education/Visualization/ Interaction/Experience | √ | √ | √ | √ |
| Demographics | √ | √ | √ | √ |
| Dimensionality/Complexity | | | | √ |

**Author Contributions:** Conceptualization, I.P. and A.T.; methodology I.P.; formal analysis, A.T. and S.K.; investigation, I.P.; writing—original draft preparation, I.P.; writing—review and editing, I.P., A.T. and S.K.; visualization, I.P.; supervision, A.T. and S.K. All authors have read and agreed to the published version of the manuscript.

**Funding:** This research received no external funding.

**Institutional Review Board Statement:** Not applicable.

**Informed Consent Statement:** Not applicable.

**Data Availability Statement:** No new data were created.

**Conflicts of Interest:** The authors declare no conflict of interest.

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
