# Peer review of "How Is Privacy Behavior Formulated? A Review of Current Research and Synthesis of Information Privacy Behavioral Factors"

_mti, doi:10.3390/mti7080076_

Round 1
Reviewer 1 Report
Very interesting and ellaborated piece of work.
A few issues that could be improved:
- A few references (more than 10 years old) could have new references obsoleting them.
- The selection of the "factors" could be better justified. In some cases it looks like they are overlapping or different things are grouped together. For example, relationship between 4.2, 4.5 and 4.8).
- Geographical location of the studies analysed is only mentioned in a few cases.
- There are some contradictions identified in the literature (such as in 4.1, 4.2 and 4.3) but explanations and conclusions are missing.
Author Response
Dear reviewer,
We sincerely appreciate your review and your constructive comments and suggestions, which helped improve the quality of the manuscript. Below we provide point-to-point responses correspondingly.
Please see the attachment
Kind Regards,
The Corresponding Author

Reviewer 2 Report
The authors present the results of an in-depth literature review on the factors affecting privacy behavior. In particular, they examine the results of several research studies to find evidence that privacy behavior is affected by various factors. Moreover, the authors synthesize a framework that aggregates the scattered factors that affect privacy behavior. Finally, the authors conclude by stating that their framework can be beneficial to academics and practitioners in the private and public sectors.
The research topic addressed by the authors is fascinating, but their work needs improvements; some suggestions are reported in the following.
Abstract
The abstract needs to be more detailed. In particular, it is difficult to understand the addressed problem and its proposed solution. Therefore, it is kindly suggested that the authors improve the abstract by better describing the problem and the presented solution. Additionally, the abstract contains many verbose sentences that make the reading difficult.
Introduction
The Introduction section needs to be improved. In particular, the storyline the authors want to define is not linear. It is kindly suggested that the authors better define the problem and describe the aim of the proposed research. Moreover, the authors never discuss user privacy awareness in a crucial context like the social network domain, which could be a breaking point for disclosing user data. To this end, in what follows I suggest some papers that the authors might consider to improve the discussion in the introduction section:
"Bhagat S, et al. Privacy-preserving user profiling with Facebook likes, IEEE International Conference on Big Data, pp. 5298--5299, 2018."
"Cerruto F, et al. Social network data analysis to highlight privacy threats in sharing data. Journal of Big Data, Vol. 9, p. 19, 2022."
"Li K, et al. Voluntary sharing and mandatory provision: Private information disclosure on social networking sites, Information Processing & Management, Vol. 57, pp. 102128, 2020."
Finally, the introduction paragraphs are not linked to each other, producing a not linear reading. It is kindly suggested that the authors harmonize the introduction section. Additionally, it is also suggested to improve the contribution paragraph by using a bullet list to describe the article's main contributions.
Related Work
The Related Work section is well categorized, but the contributions of the cited articles are challenging to understand. In this respect, It is kindly suggested that the authors improve the description of the cited works by adding pros and cons for each cited article.
Discussion
In its current form, the discussion section does not answer the research questions defined in the introduction section. In this respect, the discussion section could be modified by adding the research questions and answers to clarify the authors' contribution.
Typos and spelling English checks are required.
Author Response
Dear Reviewer,
We sincerely appreciate your review and your constructive comments and suggestions, which helped improve the quality of the manuscript.
Please see the attachment.
Kind Regards,
The Corresponding Author

Round 2
Reviewer 2 Report
The authors have addressed most of the remarks suggested. However, as a minor remark, it is kindly suggested that the authors better connect paragraphs into the introduction section and move the article’s main contributions from the Conclusion to the Introduction section. Moreover, in the following, the authors can find some typos to be fixed.
Typos
- Table 3, in its first view, contains seven columns, whereas the other ones contain eleven columns, there is a mismatch of columns concerning the views of Table 3. It is kindly suggested that the authors fix the dimensionality problem.
- Reference [73], in the References section, is empty. It is kindly suggested that the authors fix the problem.
Typos and spelling English checks are required.
Author Response
Dear Reviewer
We sincerely appreciate your review and your constructive comments and suggestions, which helped improve the quality of the manuscript. Below we provide point-to-point responses correspondingly.
The authors have addressed most of the remarks suggested. However, as a minor remark, it is kindly suggested that the authors better connect paragraphs into the introduction section and move the article’s main contributions from the Conclusion to the Introduction section.
Response: Thank you for your valuable comment. We agree with your suggestion, and in response, we moved a part of our contributions to the introduction section, as shown below, and highlighted them in our manuscript. We also made minor changes to the introduction section to make it easier to read.
Moved text:
In conclusion, this research provides the following main contributions:
- A list of all identified in the literature determinant factors of ICT users’ privacy behavior
- A framework that synthesizes and classifies the identified factors
- A fertile environment for future research to stimulate privacy protective behavior
- Inter-disciplinary perspective and favorable environment for further research opportunities in the privacy behavior domain.
Table 3, in its first view, contains seven columns, whereas the other ones contain eleven columns, there is a mismatch of columns concerning the views of Table 3. It is kindly suggested that the authors fix the dimensionality problem.
Response: Thank you for your input. It was a matter of visibility. The pages containing table 3 should be horizontal rather than vertical, while all other pages should be vertical. We corrected the layout error, and table 3 now looks correct.
Reference [73], in the References section, is empty. It is kindly suggested that the authors fix the problem.
Response: Thank you for your input. We accidentally double-entered "enter" twice while creating the references section. It is now fixed.